# Current Biomarkers in Non-Small Cell Lung Cancer—The Molecular Pathologist’s Perspective

**DOI:** 10.3390/diagnostics15050631

**Published:** 2025-03-05

**Authors:** Konrad Steinestel, Annette Arndt

**Affiliations:** Institute of Pathology and Molecular Pathology, Bundeswehrkrankenhaus Ulm, 89081 Ulm, Germany; annette2arndt@bundeswehr.org

**Keywords:** non-small cell lung cancer, biomarker, molecular pathology, next-generation sequencing

## Abstract

Non-small cell lung cancer (NSCLC) is the leading cause of cancer-related mortality worldwide. Advances in tissue-based biomarkers have significantly enhanced diagnostic and therapeutic approaches in NSCLC, enabling precision medicine strategies. This review provides a comprehensive analysis of the molecular pathologist’s practical approach to assessing NSCLC biomarkers across various specimen types (liquid biopsy, broncho–alveolar lavage, transbronchial biopsy/endobronchial ultrasound-guided biopsy, and surgical specimen), including challenges such as biological heterogeneity and preanalytical variability. We discuss the role of programmed death ligand 1 (PD-L1) immunohistochemistry in predicting immunotherapy response, the practice of histopathological tumor regression grading after neoadjuvant chemoimmunotherapy, and the application of DNA- and RNA-based techniques for detecting actionable molecular alterations. Finally, we emphasize the critical need for quality management to ensure the reliability and reproducibility of biomarker testing in NSCLC.

## 1. Introduction

Non-small cell lung cancer (NSCLC) is the leading cause of death from cancer worldwide [1]. Lung cancer represents around 10% of cancer diagnoses and 18% of cancer deaths. While there is an overall decrease in the incidence of lung cancer, reflecting the change in smoking habits, the incidence among women and non-smokers is increasing [2]. The term NSCLC summarizes lung adenocarcinoma (LAC, derived from glandular cells), lung squamous cell carcinoma (SCC, derived from metaplastic squamous epithelium), and large cell carcinoma (LCC), which may also show neuroendocrine differentiation (LCNEC) [3]. Small cell lung cancer (SCLC), on the other hand, arises from neuroendocrine cells in the bronchial or bronchiolar epithelium, which have accumulated genetic damage upon exposure to the carcinogenic components of cigarette smoke. While around 75% of lung cancers are NSCLCs in both females and males, adenocarcinomas are more frequent in women (57% vs. 39%), while squamous cell carcinomas are more frequent in men (25% vs. 12%). Small cell carcinoma and large cell carcinoma show a similar distribution between men and women (11% vs. 9% and 8% vs. 6%) [4]. There are many recent and comprehensive reviews on the epidemiology and risk factors for lung cancer, which we will therefore not discuss here; instead, the focus of this review is the (molecular) pathologist’s practical approach to the assessment of tissue-based biomarkers in NSCLC samples (cytological specimens, liquid and tissue biopsies, or resection specimens). Of note, the stage-specific impact of predictive biomarkers may change over time: while the detection of *EGFR*/*ALK* alterations has been a prerequisite for targeted therapy in late stages of lung cancer, the potential negative predictive value of these alterations for the response to neoadjuvant chemoimmunotherapy has moved these biomarkers into early/operable disease stages [5,6].

Since the development of tissue-based biomarkers and associated therapies in NSCLC is evolving so rapidly, this review can only be a snapshot of the current “state of the art” and, obviously, to a certain extent, reflects the policies and methods from the authors’ own experience. However, we will try to not only summarize our own diagnostic approach but also reflect the current literature as well as relevant guidelines in the respective sections. Taken together, the purpose of this review is to provide a concise summary of the current state of biomarker testing in NSCLC from the pathologist’s perspective.

## 2. Types of Biological Specimens in Neoplastic Lung Pathology

Tissue samples from lung neoplasms can only be obtained invasively by transthoracic (CT-guided) or transbronchial (bronchoscopic) biopsy or by a surgical procedure (atypical/wedge or anatomic resection). Therefore, as much reliable information as possible has to be obtained from the sample that reaches the pathology lab, irrespective of its size, pre-analytics, or tumor cell content. If the patient is unfit for bronchoscopy or surgery, cytology from broncho–alveolar lavage (alveolar washing) or liquid biopsy from peripheral blood may represent valuable alternatives. When considering liquid biopsy, different approaches must be discerned: the isolation of circulating tumor cells (CTCs) and the detection of cell-free (tumor) DNA (cfDNA/ctDNA) in the peripheral blood. CTCs are intact, viable cells that can offer insights into both the spatial and temporal heterogeneity of tumors, as well as their biology—areas that ctDNA mutation signatures cannot fully address. Of note, it has previously been shown that gene fusions can be reliably detected in CTCs from *ALK*-translocated NSCLC, predicting the clinical outcome [7]. Since CTCs and ctDNA each have distinct advantages and limitations, they have been proposed as complementary cancer biomarkers [8]. Table 1 summarizes the advantages and disadvantages of the respective types of biological specimens in neoplastic lung pathology. In this context, it should be noted that multidisciplinary discussion and planning of the diagnostic procedures that take into account the clinical state of the patient (lung function, comorbidities), the type and localization of the lesion (central vs. peripheral, parenchyma vs. lymph nodes), and the methodology of subsequent biomarker testing are the best approach to ensure that most biological information can be obtained from the respective tissue sample.

In patients who are unfit for any kind of invasive procedure, molecular imaging (the visualization of pathophysiological processes on a molecular level) might represent a valuable tool to obtain biological information in a non-invasive way [9]. One example of this is the correlation between radiomic texture features and *POSTN* expression levels in a preclinical model of glioblastoma [10]; another example is the correlation between FAPi PET uptake and FAP protein expression in interstitial lung disease [11,12].

**Table 1 diagnostics-15-00631-t001:** Advantages and disadvantages of types of biological specimens in neoplastic lung pathology.

Type of Specimen	Main Advantages	Main Disadvantages	Refs.
Liquid biopsy (peripheral blood)	Less invasive procedure Follow-up possible	Low negative predictive value No histopathological diagnosis	[13,14]
Broncho–alveolar lavage	Less invasive procedure High DNA quality (air-dried smears)	No tissue context Tumor cell content may be scarce IHC may be impossible/hampered	[15,16]
Transbronchial biopsy/endobronchial ultrasound-guided biopsy (EBUS)	Tissue context IHC workup possible	Invasive procedure Tumor cell content may be scarce Tumor heterogeneity might be underrepresented Histopathological grading unreliable Mimickers/pitfalls	[17,18]
Surgical specimen	Sufficient tissue for molecular workup Consideration of tumor heterogeneity Reliable histopathological grading and regression grading Extension to oncologic resection (including LN dissection) possible	Most invasive procedure (minimally invasive approaches: thoracoscopy and minithoracotomy) Some patients may be unfit for surgery	[19]

## 3. Programmed Death Ligand 1 (PD-L1)

Programmed death ligand 1 (PD-L1) is expressed on tumor cells and acts as a suppressor of the antitumoral immune response through interaction with PD-1, which is expressed on immune cells [20,21]. These would normally recognize and attack neoantigen-presenting tumor cells. This can be therapeutically exploited by inhibiting PD-L1 through PD-L1-binding antibodies, thus giving the patient’s immune system the chance to recognize and destroy the malignant cells. The effectiveness of pharmaceutical PD-L1 inhibition depends on the amount of PD-L1 on the surface of tumor cells, with PD-L1-positive tumors showing a better response to (chemo)immunotherapy compared to PD-L1-negative tumors. PD-L1 expression in tumor cells is routinely assessed by PD-L1 immunohistochemistry (IHC), for which different antibody clones and protocols are used by pathologists. Some of these are commercially available assays (SP142, SP263, 28-8, 22C3), while others are laboratory-developed tests (E1L3N) [22]. In general, large comparability studies have shown that these assays provide both comparable and reliable results, given that measures of quality assurance (e.g., ring trials) are adequately employed and rigorously followed [23,24]. Of note, all caveats that apply to the use of IHC, in general (pre-analytics, over-/underfixation, heat treatment, buffers, dilution), as well as the respective antibody clone and staining platform, should be taken into account when interpreting PD-L1 staining. External quality assurance as well as continuous monitoring can help to standardize the methodology between different laboratories [25]. However, PD-L1 staining intensities may not only vary depending on methodology but also for biological reasons. PD-L1 expression in immune cells represents IFN-γ-induced adaptive regulation and is associated with an increase in tumor-infiltrating lymphocytes and effector T cells, while high PD-L1 expression on tumor cells has been linked to epigenetic dysregulation of the PD-L1 gene [26]. Since chemotherapy reduces PD-L1 expression in tumor cells for a subset of patients [27], rebiopsy and reassessment of PD-L1 expression may be necessary to determine eligibility for immune checkpoint inhibitor therapy, but so far, there is no standardized PD-L1 testing strategy. Such a guideline should include optimal reassessment time points, the required number of biopsies, and the evaluation of surgical specimens [28].

The PD-L1 tumor proportion score (TPS) is the ratio between PD-L1-positive tumor cells and all tumor cells in the respective sample; high TP scores are associated with better response to immune checkpoint inhibitors targeting PD-1 or PD-L1 and higher overall survival in NSCLC patients [29,30]. However, the evaluation of PD-L1 staining is challenging due to biological heterogeneity, slightly different performance of antibody clones, and relevant interobserver variation [31,32]. A weighted kappa for interobserver variation between pathologists of 0.71–0.96 when assessing TPS in NSCLC has been described [33]; in the same study, up to 20% of the cases showed discordant classification as positive or negative using TPS ≥ 1% as the cutoff (0–5% when using a cutoff of TPS ≥ 50%). Other studies confirmed relatively high agreement while suggesting that training in predefined areas could improve reproducibility [34]. It has been highlighted that distinguishing “true positive” from “false-positive” artifacts can be difficult, especially in specimens with lower percentages of positive cells and faint staining [35]. This is of utmost clinical relevance since under- or overscoring might result in under- or overtreatment of NSCLC patients.

Artificial intelligence (AI)-assisted TPS scoring in NSCLC has been shown to be feasible, with results comparable to the assessment by experienced pathologists or even outperforming them [36,37]. The reproducibility and efficiency of untrained pathologists could also be improved by AI assistance [38]. However, the use of AI still does not overcome the major challenges in TPS scoring: first, traditional AI approaches rely solely on real patient data. As a result, cases that are highly relevant for training (e.g., those with a TPS of around 1% or 50%) follow a natural biological distribution and may be under-represented in training cohorts, which typically include no more than a few hundred cases. Second, the real-world training cases must be carefully annotated by real pathologists, raising the possibility of perpetuating the above-mentioned biases during AI training. There are recent alternatives to IHC when assessing PD-L1 expression: Soluble PD-1 (sPD-1) and soluble PD-L1 (sPD-L1) are present in the bloodstream as free proteins and can be quantified using enzyme-linked immunosorbent assays (ELISA). However, ELISA relies on monoclonal antibodies, which are expensive to produce and require significant time for isolation and purification [39]. PD-L1 exosomes can be detected by optical technologies such as Surface Plasmon Resonance (SPR) spectroscopy [40]. Regarding non-tissue-based methods, [^18^F]DK222-PET has been proposed as a non-invasive imaging tool to detect variable PD-L1 expression in tumors [41].

## 4. Assessment of Tumor Regression

With the approval of neoadjuvant chemoimmunotherapy in NSCLC, it is the pathologists’ task to assess the tumor response to therapy when evaluating the lung resection specimen. There are two main schemes for regression grading after neoadjuvant therapy in NSCLC: the grading system by Junker et al. has originally been published in 1997 to assess the response to neoadjuvant radiochemotherapy in NSCLC [42]; in 2020, the IALSC published an expert consensus for the handling and examination of NSCLC specimens after all neoadjuvant treatment schemes, including targeted therapy and immunotherapy [43]. Complete tumor regression (no residual viable tumor, corresponding to regression grade III (RGIII) in the Junker scheme or pathological complete response (pCR, Figure 1a) in the IASLC scheme) has been shown to predict event-free survival in the CheckMate 816 trial, qualifying tumor regression grading upon neoadjuvant chemoimmunotherapy as a prognostic biomarker in NSCLC [44]. The association between pCR/major pathological response (MPR in the IASLC scheme, RG IIb in the Junker scheme; Figure 1b) and EFS has been confirmed in a recent meta-analysis; however, a significant correlation between pCR/MPR and overall survival could not yet be proven [45]. To enhance robustness of tumor regression as a prognostic biomarker and as an endpoint for the identification of novel predictive biomarkers for the effectivity of chemoimmunotherapy, it is crucial that pathologists perform regression grading in a comprehensive and standardized way, including thorough macroscopic assessment of the resection specimen, embedding of the whole tumor bed (or a representative slide of the largest diameter), examining all lymph nodes, and reporting the percentage of residual vital tumor (% RVT) as a continuous variable. We currently conduct a multicenter study (Re-GraDE Germany) to evaluate the current state of the art of tumor regression grading in NSCLC specimens after neoadjuvant chemoimmunotherapy in Germany (manuscript in preparation).

## 5. DNA- and RNA-Based Biomarkers

Tumor-promoting molecular alterations not only underlie tumor formation and progression in NSCLC but also represent targets for personalized treatment [46]. These include driver mutations (e.g., *EGFR*, *BRAF*, *KRAS*, *MET*), translocations/fusions (e.g., *ALK*, *ROS*, *RET*), and gene amplifications (e.g., *ERBB2*, *FGFR1*). The ESCAT classification (ESMO Scale for clinical actionability of molecular targets) classifies biomarkers with approved targeted therapies (*EGFR*, *ALK*, *ROS*) into category I (A, B, C) [47]. While there are comprehensive reviews on the fast-growing list of targeted therapy options for each alteration and while the focus of this review lies on the detection of these alterations in different tissue samples, we will shortly summarize the current knowledge on each mutation. Of note, the distribution of targetable genetic alterations differs significantly between the histologic subtypes of NSCLC. While targetable driver mutations (e.g., *EGFR*, *BRAF*) and translocations/fusions (e.g., *ALK*, *ROS*, *RET*) are frequently detected in lung adenocarcinoma (see next paragraph), they are rare (<1%) in squamous cell carcinoma and neuroendocrine neoplasms. Only KRAS G12 mutations and *MET* amplifications/*MET* exon 14 skipping mutations are observed in 2.1 and 1.5% of lung squamous cell carcinoma, respectively [48]. Notably, the comparable frequency of KRAS G12C mutations in the “large cell carcinoma” histology group and lung adenocarcinoma supports the hypothesis that a substantial portion of large cell carcinomas are undifferentiated TTF1-negative adenocarcinomas.

Mutations in epithelial growth factor receptor (*EGFR*) can be detected in about 15% of NSCLC in Europe and America and in up to 50% of cases in Asia [49]. They are more frequent in lung adenocarcinomas, in women, and in non-smokers. Tyrosine kinase inhibition (TKI)-sensitizing mutations occur in *EGFR* exons 18–21, with in-frame exon 19 deletions and the exon 21 point mutation L858R (also considered “classical” *EGFR* mutations) constituting over 90% of all *EGFR* mutations [50,51]. Of note, rarer *EGFR* mutations (e.g., exon 20 insertions) are associated with decreased sensitivity to TKI treatment [52]. Targeted therapies targeting EGFR comprise antibodies against the extracellular domain that block the dimerization of the receptor or small molecule tyrosine kinase inhibitors, which bind to EGFR and block signal transduction [53]. The increased ATP-binding affinity of the mutant EGFR protein due to conformational change increases TKI binding and suppresses downstream signaling [50]. There is conflicting evidence on the prognostic role of *EGFR* mutations; while earlier studies found no differences in prognosis between *EGFR* wild-type and *EGFR*-mutant cases, advances in EGFR-targeting tyrosine kinase inhibition have led to improved survival in patients whose tumor harbors an *EGFR* alteration [54]. There are reports indicating a more aggressive biological behavior of tumors with *EGFR* exon 19 deletion compared to tumors harboring the EGFR L858R mutation [55]. Like other oncogenes, *EGFR* mutation is associated with a higher rate of metastatic disease in the central nervous system compared to non-oncogene-addicted NSCLC [56].

In general, *EGFR* mutations are detected in tumor tissue or liquid biopsies/cfDNA by DNA sequencing, but PCR-based approaches for the targeted detection of specific hotspot mutations also exist. For example, it has been shown that the Amplification Refractory Mutation System (ARMS) that selectively amplifies mutation-containing target sequences has a higher sensitivity compared to tissue-based DNA sequencing [57]; however, additional mutations are not detected by this approach. The same is true for fragment length analysis and pyrosequencing, the latter approach also being limited by the requirement of a relatively high tumor cell fraction (>20%) in the investigated sample. In addition, there are mutation-specific antibodies against exon 19-deleted or L858R-mutant EGFR, but given the multitude of immunohistochemical, DNA-, and RNA-based biomarkers, which have to be evaluated in a limited NSCLC tissue sample, we would abstain from the use of tissue slides for targeted detection of individual mutations. This limitation may one day be overcome by multiplex immunostaining.

V-Raf Murine Sarcoma Viral Oncogene Homolog B (*BRAF*) mutations, which mostly affect the activation loop (A-loop) around codon 600, are detected in 3–8% of NSCLC cases [58]. In NSCLC, class 1 (V600E, activity independent from upstream RAS signaling), class 2 (non-V600E, functionally active as a dimer, with intrinsic discrete kinase activity, independent from upstream RAS signaling), and class 3 mutations (non-V600E, no enzymatic activity, dependent on upstream RAS stimulation) are evenly distributed. This is in contrast to malignant melanoma where the vast majority of *BRAF*-mutant tumors harbor V600E (class 1) mutations. Most patients with *BRAF*-mutant NSCLC are former or current smokers; however, up to 30% have never smoked [59]. Comparable to *EGFR* mutations, *BRAF* mutations are associated with a higher rate of metastatic disease to the CNS in NSCLC [60]. While tyrosine kinase inhibition is effective, especially in class 1 (V600E) mutations, patients will eventually acquire additional mutations and develop resistance to therapy [61]. Moreover, treatment of class 2 and 3 mutations is not equally effective [58]. Similar to *EGFR*, the method of choice for the detection of *BRAF* mutations is DNA sequencing, mostly with NGS, which can also be applied to liquid biopsies. Given the even distribution of V600E and non-V600E *BRAF* mutations in NSCLC, it is mandatory that fast-track/targeted sequencing approaches span the entire range of possible mutations.

Kirsten rat sarcoma (*KRAS*) mutations are the most frequent oncogenic driver alterations in NSCLC, detectable in up to 30% of cases in Caucasian patients, with much lower rates in patients of Asian descent [62]. Activating mutations are associated with smoking history and mostly affect codons 12 (90%) or 13, thus keeping the mutant small GTPase KRAS in a constitutionally active state. They frequently occur together with co-mutations in *TP53*, *STK11*, or *KEAP1* (see below) [63]. While *KRAS* mutations have long been deemed undruggable, the recent approval of KRAS G12C-specific inhibitors has proven the principle of effective targeted therapy in that setting [62]. Of note, secondary mutations in *KRAS* or co-mutations in other genes can lead to therapy resistance. With respect to immunotherapy, *KRAS*-mutant NSCLC seems to experience higher responses to immune checkpoint inhibition compared to other molecular drivers [64]. The prognostic role of *KRAS* mutations in NSCLC is still a matter of debate due to conflicting results in the literature [62]. Regarding the methodology of detection, the relatively restricted localization of the most frequent *KRAS* mutations to certain hotspots (codons 12/13, 61) allows for targeted detection (e.g., PCR-based techniques) [65]. In NSCLC, however, the necessity for comprehensive characterization (including other alterations) will lead to the use of NGS in most cases, including liquid biopsies.

Alterations in the *MET* oncogene (up to 4% of NSCLC) include MET protein overexpression, mutations leading to *MET* exon 14 skipping, or *MET* gene amplification [66]. Not only do these occur as primary driver mutations but also as resistance-mediating secondary mutations upon targeted treatment of oncogene-addicted NSCLC. METex14 mutations are in most cases detected by NGS, with hybrid capture-based panels showing higher sensitivity compared to amplicon-based panels; of note, a DNA-based approach only will not cover the entire spectrum of possible METex14 mutations [66]. MET protein overexpression can be detected by immunohistochemistry, while *MET* gene amplification will be covered by fluorescence in situ hybridization (FISH) [67].

Alterations in the anaplastic lymphoma kinase (*ALK*) oncogene are found in around 5% of NSCLC cases and include gene fusions, gene amplifications, and activating point mutations [68]. While there are more than 20 different described fusion partners for *ALK*, the most frequent fusion in NSCLC is the one between the 3′ region of the *ALK* gene and the 5′ region of the echinoderm microtubule-associated protein-like 4 (*EML4*) gene [69]. The fusion enhances ALK activity with subsequent signaling along pro-mitogenic pathways such as the mitogen-activated protein kinase (MAPK), (phosphatidylinositol 3−kinase) PI3K/(protein kinase B) AKT, Janus kinase/signal transducer and activator of transcription (JAK/STAT), and mitogen-activated protein kinase 5–extracellular signal-regulated kinase 5 (MEK5-ERK5) pathways [68]. The diagnosis of *ALK* rearrangements can be performed by using immunohistochemistry (relying on the upregulation of membranous ALK protein expression upon gene fusion), FISH (by directly visualizing the breaking apart of the *ALK* gene, with or without probing the suspected fusion partner), or RNA-based NGS. As stated above, each technique has its own advantages and disadvantages and the selection of the method depends on the type and amount of available tissue, the expected turnaround time, and, of course, the local availability and reimbursement regulation of the respective technique. Some laboratories will verify a positive ALK IHC by FISH testing, leading to discordant results in rare cases. For these, it has been shown that patients who were *ALK* FISH-negative but IHC-positive show a response to ALK-targeted therapy, giving ALK IHC a higher predictive value from a clinical viewpoint [70]. It is possible to detect *ALK* alterations in liquid biopsies (especially in circulating tumor cells); however, there are some technical hurdles to cfDNA/plasma-based detection by NGS given the rapid RNA degradation in blood samples [71].

Proto-oncogene tyrosine-protein kinase-1 (*ROS1*; c-Ros oncogene-1)-gene fusions occur in up to 2% of NSCLCs, associated with female gender, non-smokers, and adenocarcinoma histology [72,73]. While the physiologic role of the protein is not yet fully clear, *ROS1* seems to play a key role in the embryonic development of epithelial tissue [74]. Upon an oncogenic microdeletion, *ROS1* fuses with fused-in-glioblastoma (*FIG*), leading to ROS1 overexpression and the activation of downstream signaling pathways [75]. This can be exploited by the immunohistochemical detection of oncogenic ROS1 in NSCLC tumor cells [76]; however, recent data have questioned the specificity of the available ROS1 antibodies and advocated for the use of reflex NGS for the detection of *ROS1* alterations [77]. Finally, *RET* (rearranged during transfection) or *NTRK* (neurotrophic tropomyosin-receptor kinase) alterations that are druggable but comparably rare in NSCLC can be proven by FISH or RNA-based NGS, with panTRK IHC being used as a tissue-agnostic screening marker for *NTRK* alterations [78].

## 6. The Role of Co-Mutations

In recent years, there is growing evidence that not only does the presence of a single driver mutation but also of additional/secondary mutations affect the biological behavior, as well as the response to targeted therapy in NSCLC, a phenomenon called “intra-driver heterogeneity” [79]. Co-mutations in *STK11*, *KEAP1*, and *TP53* are frequently observed in *KRAS*-mutant lung adenocarcinoma, giving rise to a more aggressive tumor phenotype and representing an independent negative prognostic indicator. In *EGFR*-mutant lung adenocarcinoma, more than half of the cases harbor mutations in *TP53*, allowing for greater genetic instability and a larger burden of mutations, which may confer resistance to TKI inhibitors [80]. Of note, *TP53* mutations are underrepresented in the group of NSCLC associated with gene fusions (*ALK*, *ROS*).

## 7. Genome-Wide Biomarkers: Tumor Mutational Burden and Microsatellite Instability

In addition to alterations in single genes or gene fusions, genome-wide biomarkers such as tumor mutational burden (TMB) and microsatellite instability have been investigated as possible predictive biomarkers for the response to immune checkpoint inhibition. Tumor mutational burden is defined as the number of somatic mutations per megabase and shows great variability between identical tumor entities from different patients as well as between different tumor entities [81]. Tumors with high TMB produce and display a high number of neoantigens, which are then recognized by the immune system, especially under checkpoint inhibitor treatment. Of note, TMB can not only be assessed in tissue samples but also in liquid biopsies, and thus would represent a valuable biomarker that could also be obtained from frail patients at minimal risk [82]. However, the predictive value of TMB for ICI efficiency has so far not been consistently proven [83]. While originally TMB had to be assessed by whole exome sequencing, it has been shown that the results can reliably be obtained by large panel sequencing/comprehensive genomic profiling [81].

Microsatellite instability (MSI-H) and mismatch repair deficiency (dMMR) describe an oncogenic process where somatic or germline mutations in MMR genes lead to an increase in mutations during DNA replication, resulting in high mutational load and tumorigenesis. While rather frequent in colorectal and endometrial carcinoma, MSI-H/dMMR is rare in NSCLC (<1%) and is mostly associated with smoking and adenocarcinoma histology [84]. These tumors show high TMB and are in general vulnerable to ICI treatment, while co-occurring mutations in *STK11* and *KEAP1* seem to be associated with poor response. While dMMR cases can be identified by IHC for MMR proteins (MLH1, MSH2, MSH6, PMS2), this might be limited by the amount of available tissue, especially given the low rate of NSCLC cases in which dMMR can be expected. A possible solution lies in the use of larger NGS panels, which are capable of detecting both TMB and MSI status in parallel to individual mutation detection.

## 8. Emerging Biomarkers

In addition to single-gene and genome-wide biomarkers, there are several emerging biomarkers that are currently under investigation in NSCLC, some of which require additional methodological considerations from the (molecular) pathologist. Proteins that are involved in the regulation of the cell cycle (such as p16) and proliferation markers (such as Ki67) that can be assessed with immunohistochemistry have shown promising results as prognostic biomarkers [85]. Antibody–drug conjugates (ADCs) target tumor cell (surface) antigens and deliver a cytotoxic drug load to the tumor cell [86]. The aim of this approach is to reduce unspecific cytotoxic effects that contribute to the toxicity of the treatment, but so far, no ADC has been approved for the treatment of NSCLC. The expression of the respective cellular targets (Her2, Her3, Trop2, Nectin4, MET) is assessed with immunohistochemistry, giving rise to a new group of IHC-based predictive biomarkers for which quality-controlled and reproducible assessment is mandatory. However, the possible requirement to assess a multitude of novel IHC-based biomarkers on a very limited tissue sample should speed up the implementation of multiplex immunohistochemistry so that multiple markers can be assessed on a single slide [87].

## 9. Technical Aspects

Taken together, as shown in Table 1, (molecular) pathologists encounter a variety of samples, each sample type harboring individual advantages and disadvantages for biomarker testing in NSCLC. The current ESMO guidelines advocate for molecular testing in all lung adenocarcinomas and squamous cell carcinomas in young patients/non-smokers and include the possibility of tumor genetic testing in liquid biopsies, although these are not (yet) regarded as equivalent to tissue [46]. With respect to the recommended method, next-generation sequencing (NGS) has evolved to be the “workhorse” of molecular lung pathology [88]. Both amplicon- and hybrid capture-based techniques are capable of detecting DNA alterations (mutations, copy number alterations) in a large number of samples in a relatively quick laboratory turnaround time [89]. Large NGS panels (comprehensive genomic profiling) are capable of detecting genome-wide biomarkers such as tumor mutational burden and microsatellite instability. Smaller NGS panels are unable to detect genome-wide biomarkers and might not include genes with an emerging role as prognostic and/or predictive biomarkers such as *STK11* or *KEAP1* [90].

The limitations of NGS testing lie in the amount of sample tumor DNA (>40 ng), which is necessary, especially for hybrid capture-based sequencing techniques as well as in the laboratory turnaround time [91]. In addition, fusions/translocations with unknown partners cannot be detected by DNA-based NGS. RNA-based NGS or fluorescence in situ hybridization (FISH) can be used in addition, with the latter technique requiring additional unstained slides from often very limited material. For *ALK* and *ROS* translocations, immunohistochemistry is another alternative, but recent data support the use of RNA NGS due to higher speed and comparable reliability for the detection of *ALK* fusions and higher reliability for the detection of *ROS* fusions [77,92]. When performing RNA NGS, however, one has to take into account formalin fixation artifacts and the lower stability of RNA, leading to RNA degradation [93]. With the widespread use of NGS, subsequent testing of individual genes is discouraged since it has been shown that NGS is not only more comprehensive but also more cost-effective compared to single-gene testing [94]. Single-gene testing, however, may have a certain role in a “fast-track” setting when only individual mutations must be ruled out before starting immediate therapy, and the turnaround time for full-scale NGS would be too long. Of note, most of the aspects discussed here concern the situation in countries where NGS technology is available. In countries where a significant number of patients and pathologists do not have access to these technologies, alternative methods of biomarker detection (immunohistochemistry or PCR-based testing) have to be considered. A 2020 IASLC survey among 2537 respondents from 102 countries revealed dissatisfaction with the current state of molecular testing for lung cancer in their respective countries, and of note, 33% of health care professionals were unaware of the current guidelines for molecular testing issued by the College of American Pathologists, IASLC, and Association for Molecular Pathology [95]. Beyond financial and methodological constraints in many regions, these findings underscore the critical need for continuous knowledge dissemination and best practice guidelines on biomarker testing in NSCLC.

## 10. Quality Management in Biomarker Testing and Outlook

For all discussed biomarkers, it is of utmost importance that (molecular) pathologists make sure that preanalytical requirements are met and that (at least internally) validated tests are used thoroughly. Participation in national or international interlaboratory ring trials should be mandatory [96,97,98]. Correlations between clinical, histopathological, and molecular tumor characteristics, as well as taking into consideration tissue preservation and tumor cell type and content, make the interpretation of the results from biomarker testing both more straightforward and reliable. From our point of view, the outsourcing of (molecular) biomarker tests from pathology institutes as well as the separation of morphological and molecular evaluation of biomarkers, is extremely critical. Instead, we strongly support extended molecular pathology training, as is exemplified by the introduction of the comprehensive Master’s degree program called the “European Masters in Molecular Pathology” (EMMP) by the Pathology Section of the European Union of Medical Specialists and the European Society of Pathology [99]. Well-trained molecular pathologists will be extremely valuable participants in interdisciplinary/molecular tumor boards since they will be able to communicate the pros and cons as well as the limitations of the requested biomarker assay, to interpret them, and to merge the results with morphologic and clinical data. Strengthening the network between (molecular) pathologists throughout Europe and the world will assure continuous high-quality biomarker testing and form the basis for the discovery of novel biomarkers in NSCLC and beyond.

## Figures and Tables

**Figure 1 diagnostics-15-00631-f001:**
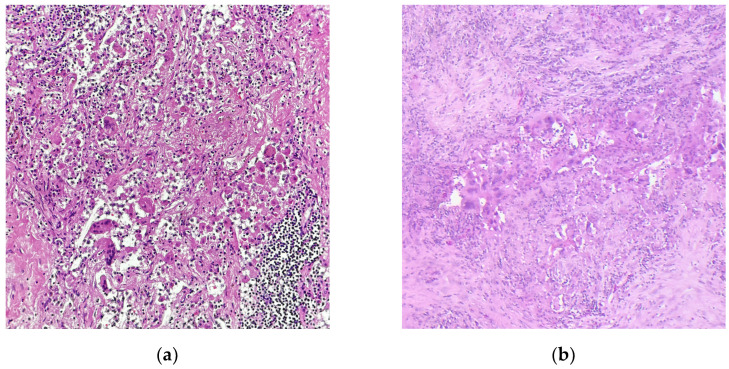
Representative microphotographs of NSCLC samples after neoadjuvant chemoimmunotherapy. (**a**) Complete pathological response (pCR); (**b**) major pathological response (MPR) with residual vital tumor cells. Hematoxylin-Eosin staining; Magnification, 200×; images courtesy of Konrad Steinestel from his personal collection.

## Data Availability

Data are available from the authors upon request.

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
