# Peer review of "Current Biomarkers in Non-Small Cell Lung Cancer—The Molecular Pathologist’s Perspective"

_diagnostics, 2025, doi:10.3390/diagnostics15050631_

Round 1
Reviewer 1 Report
Comments and Suggestions for Authors
Please, include updated epidemiological data on NSCLC and SCLC.
The purpose of the review must be made explicit.
Among the diagnostic techniques, thoracoscopy and minithoracotomy should be mentioned. in addition, the concept of liquid biopsy should be developed, mentioning circulating tumor cells and circulating genetic material. Mention any alternatives to immunohistochemical analysis for the detection of PD-L1. It is necessary to mention the factors related to the proliferative index of cells. co-mutations should be put in a separate chapter.
multidisciplinary approach should be emphasized
also should be mentioned the frequency of mutations in squamous histotype compared to adenocarcinoma.
I suggest to include the following references for the discussion
Transl Lung Cancer Res. 2021 Jan;10(1):80-92.
Anticancer Res. 2020 Feb;40(2):983-990.
Author Response
Reviewers comments:
Please, include updated epidemiological data on NSCLC and SCLC.
The purpose of the review must be made explicit.
Among the diagnostic techniques, thoracoscopy and minithoracotomy should be mentioned. in addition, the concept of liquid biopsy should be developed, mentioning circulating tumor cells and circulating genetic material. Mention any alternatives to immunohistochemical analysis for the detection of PD-L1. It is necessary to mention the factors related to the proliferative index of cells. co-mutations should be put in a separate chapter.
multidisciplinary approach should be emphasized
also should be mentioned the frequency of mutations in squamous histotype compared to adenocarcinoma.
I suggest to include the following references for the discussion
Transl Lung Cancer Res. 2021 Jan;10(1):80-92.
Anticancer Res. 2020 Feb;40(2):983-990.
Response: We would like to thank the reviewer for his evaluation of our work and his helpful comments. We have made all changes accordingly, have added more and updated references and included the recommended references.
Reviewer 2 Report
Comments and Suggestions for Authors
This article is a review of non-small cell lung cancer (NSCLC) biomarkers, comprehensively analysing the practice of NSCLC biomarker detection in different types of specimens. The article is informative, covering many aspects of NSCLC biomarker detection, and is of some reference value for research and practice in related fields.
- Figure 1 does not provide a scale or description of the staining method; additional technical details are needed.
- Are the conclusions in this paper based on the healthcare systems of developed countries? Have the differences in technology, costs and reimbursement policies in low- and middle-income countries been taken into account?
- In Programmed death ligand 1 (PD-L1) part: Standardisation and dynamic monitoring of PD-L1 assessment: the paper mentions the consistency of testing of different PD-L1 antibodies, but does not discuss whether differences in testing procedures (e.g., staining platforms, scoring thresholds) between laboratories may affect the comparability of results. PD-L1 expression may change with treatment or over time, but there is no discussion of how to address differences in PD-L1 detection in pre- and post-treatment samples (e.g., the need for re-biopsy after neoadjuvant therapy).
- It is recommended that a additional part of molecular imaging be added, as there is now a concept of ‘transparent pathology’, and molecular imaging, as a non-invasive, reproducible, and novel technique, should not be absent from this review. Please refer to doi: 10.1007/s00259-021-05234-1.
Author Response
Comment:
This article is a review of non-small cell lung cancer (NSCLC) biomarkers, comprehensively analysing the practice of NSCLC biomarker detection in different types of specimens. The article is informative, covering many aspects of NSCLC biomarker detection, and is of some reference value for research and practice in related fields.
- Figure 1 does not provide a scale or description of the staining method; additional technical details are needed.
- Are the conclusions in this paper based on the healthcare systems of developed countries? Have the differences in technology, costs and reimbursement policies in low- and middle-income countries been taken into account?
- In Programmed death ligand 1 (PD-L1) part: Standardisation and dynamic monitoring of PD-L1 assessment: the paper mentions the consistency of testing of different PD-L1 antibodies, but does not discuss whether differences in testing procedures (e.g., staining platforms, scoring thresholds) between laboratories may affect the comparability of results. PD-L1 expression may change with treatment or over time, but there is no discussion of how to address differences in PD-L1 detection in pre- and post-treatment samples (e.g., the need for re-biopsy after neoadjuvant therapy).
- It is recommended that a additional part of molecular imaging be added, as there is now a concept of ‘transparent pathology’, and molecular imaging, as a non-invasive, reproducible, and novel technique, should not be absent from this review. Please refer to doi: 10.1007/s00259-021-05234-1.
Response: We would like to thank the reviewer for his valuable and helpful comments. We have included all additional points in the revised manuscript and updated the references, including the recommended literature.